# Enhanced Biosafety of the *Sleeping Beauty* Transposon System by Using mRNA as Source of Transposase to Efficiently and Stably Transfect Retinal Pigment Epithelial Cells

**DOI:** 10.3390/biom13040658

**Published:** 2023-04-07

**Authors:** Nina Harmening, Sandra Johnen, Zsuzsanna Izsvák, Zoltan Ivics, Martina Kropp, Thais Bascuas, Peter Walter, Andreas Kreis, Bojan Pajic, Gabriele Thumann

**Affiliations:** 1Experimental Ophthalmology, University of Geneva, 1205 Geneva, Switzerland; 2Department of Ophthalmology, University Hospitals of Geneva, 1205 Geneva, Switzerland; 3Department of Ophthalmology, University Hospital RWTH Aachen, 52074 Aachen, Germany; 4Max Delbrück Center for Molecular Medicine in the Helmholtz Association, 13125 Berlin, Germany; 5Division of Medical Biotechnology, Paul-Ehrlich-Institute, 63225 Langen, Germany; 6Eye Clinic ORASIS, Swiss Eye Research Foundation, 5734 Reinach, Switzerland; 7Faculty of Sciences, Department of Physics, University of Novi Sad, Trg Dositeja Obradovica 4, 21000 Novi Sad, Serbia; 8Faculty of Medicine of the Military Medical Academy, University of Defense, 11000 Belgrade, Serbia

**Keywords:** nvAMD, PEDF, RPE cells, gene therapy, non-viral, *Sleeping Beauty* transposase, mRNA

## Abstract

Neovascular age-related macular degeneration (nvAMD) is characterized by choroidal neovascularization (CNV), which leads to retinal pigment epithelial (RPE) cell and photoreceptor degeneration and blindness if untreated. Since blood vessel growth is mediated by endothelial cell growth factors, including vascular endothelial growth factor (VEGF), treatment consists of repeated, often monthly, intravitreal injections of anti-angiogenic biopharmaceuticals. Frequent injections are costly and present logistic difficulties; therefore, our laboratories are developing a cell-based gene therapy based on autologous RPE cells transfected ex vivo with the pigment epithelium derived factor (PEDF), which is the most potent natural antagonist of VEGF. Gene delivery and long-term expression of the transgene are enabled by the use of the non-viral *Sleeping Beauty* (SB100X) transposon system that is introduced into the cells by electroporation. The transposase may have a cytotoxic effect and a low risk of remobilization of the transposon if supplied in the form of DNA. Here, we investigated the use of the SB100X transposase delivered as mRNA and showed that ARPE-19 cells as well as primary human RPE cells were successfully transfected with the Venus or the *PEDF* gene, followed by stable transgene expression. In human RPE cells, secretion of recombinant PEDF could be detected in cell culture up to one year. Non-viral ex vivo transfection using SB100X-mRNA in combination with electroporation increases the biosafety of our gene therapeutic approach to treat nvAMD while ensuring high transfection efficiency and long-term transgene expression in RPE cells.

## 1. Introduction

Additive gene therapy is a promising therapeutic strategy for diseases that require continuous or frequent administration of biopharmaceuticals as well as for polygenetic multifactorial diseases for which no effective treatment exists. Neovascular age-related macular degeneration (nvAMD), a devastating neurodegenerative disease of poorly understood etiology, is characterized by choroidal blood vessels that invade the subretinal space, impairing the function of the retinal pigment epithelium (RPE) and eventually photoreceptor function. The choroidal neovascularization (CNV) is a result of an imbalance between pro-angiogenic factors, such as the vascular endothelial growth factor (VEGF), and anti-angiogenic factors, such as the pigment epithelium-derived factor (PEDF), leading to visual impairment and potentially rapid loss of vision [1,2,3,4,5,6]. The current treatment is the intraocular injection of costly biopharmaceuticals, i.e., ranibizumab, aflibercept, brolucizumab, and bevacizumab (off-label use), which counteracts the promotion of blood vessel growth by VEGF; these biopharmaceuticals halt or retard the progression of neovascularization in 90% of patients, with 30–40% regaining some lost vision [7,8,9,10,11,12,13]. In addition to being costly, the treatment, which requires frequent, often monthly intraocular injections due to a short half-life time of a few days [14,15], is challenging for the elderly and visually impaired patients and their accompanying care-givers, and it is associated with various risks for side effects, such as endophthalmitis, cataract formation, and ocular hemorrhage, which accumulate with the repeated injections [16,17,18,19,20].

To overcome the difficulties of the current treatment for nvAMD patients, our laboratories are developing a cell-based gene therapeutic strategy for a one-time treatment. Specifically, autologous iris pigment epithelial (IPE) or RPE cells are isolated from a patients’ biopsy, transfected with the *PEDF* gene, and transplanted into the subretinal space of the same patient without cultivation and cell expansion. In the subretinal space, the transfected cells will overexpress and secrete recombinant (r)PEDF, counteracting neovascularization over a long period. The feasibility of this strategy has been shown in animal models of choroidal and corneal neovascularization [21,22,23,24]. VEGF-induced sprouting and migration of human umbilical vein endothelial cells (HUVEC) was inhibited by rPEDF with a 2000 times higher potency compared with bevacizumab [21]. We recently established the GMP-compliant production of PEDF-overexpressing IPE cells to evaluate the safety of this treatment approach in a first-in-human phase 1b/2a clinical trial [25].

Efficient and safe gene delivery to cells is essential for the development of gene therapeutic strategies based on the addition, replacement, or repair of genes to treat a number of health disorders. During the last decades, several techniques have been developed to deliver genes to cells in vivo and in vitro. Three major categories can be distinguished, namely physical, chemical, and biological (virally mediated) techniques [26,27,28,29,30,31,32]: Physical techniques, such as microinjection and electroporation of naked DNA, and chemical techniques, such as lipofection and nanoparticles, are quite safe and could be developed for clinical application, but are usually associated with low transfection efficiencies and transient gene expression. Viral vector-mediated techniques rely on modified viruses to carry the gene of interest (GOI) into cells, where it is expressed either episomally or integrated into the genome of the host cell, depending on the vector. In the last years, significant clinical successes have been reported, such as the approval by the FDA in 2017 of voretigene neparvovec-rzyl (Luxturna) to treat inherited retinal dystrophy [33,34], which is now in clinical use [35,36]. For nvAMD, several clinical trials are ongoing which are investigating the use of viral vectors to deliver genes encoding anti-VEGF molecules to the eye [37,38,39,40,41,42]. On the other hand, viral vectors suffer from a number of limitations. Amongst others, adenoviral or adeno-associated viral (AAVs) vectors deliver genes episomally, meaning that with time, the transgene is lost in dividing cell populations, requiring repeated administration to maintain the desired level of transgene expression. Moreover, adenoviral and adeno-associated viral vectors can induce harmful immune responses [43,44]. Retroviral and lentiviral vectors integrate the transgene into the host cells’ genome and, thus, have the potential for long term expression of the transgene [45]. However, as these vectors integrate preferentially around transcription start sites, in transcriptional regulatory elements or transcribed sequences, they can have severe side effects, including genotoxicity and mutagenic effects [46,47,48,49,50].

To avoid the possible complications associated with viral gene delivery, we have developed a transfection protocol using the hyperactive *Sleeping Beauty* (SB100X) transposon system and showed highly efficient transgene integration into the genome of primary human iris pigment epithelial (IPE) and RPE cells and high expression and secretion of rPEDF [51,52,53,54,55,56,57]. Our strategy of gene delivery using the non-viral SB100X transposon system and electroporation combines the advantages of viral and physical delivery methods [58,59,60,61], namely high cargo capacity, high gene transfer rates, high gene expression rates, and long-term expression of the transgene, which is integrated into the host cells’ genome randomly into TA motifs, thus having a minimal risk for insertional mutagenesis without triggering an immune response [62,63,64,65]. The system consists of two plasmids, one encoding the hyperactive SB100X transposase, and a second plasmid carrying the GOI flanked by terminal inverted repeats (TIR) of the transposon [51,66,67,68]. Once the plasmids are delivered into the cells by electroporation, the transposase recognizes the TIRs, cuts out the GOI/transposon from the vector plasmid and inserts it randomly into the genome of the host cells (Figure 1).

Even though the *Sleeping Beauty* transposon system is known to be safe and no harmful side-effects have been reported thus far, the transposase coding sequence bears a low risk of spontaneous integration if supplied as DNA, which would result in a continuous expression of the transposase and possible remobilization of the transposon, consequently increasing the risk of insertional mutagenesis [69,70,71]. The sequence of the transposase gene was detected in the genome of HEK293 cells transfected with SB100X DNA [72]. Huang et al. showed random integration of the *Sleeping Beauty* sequence into the genome of T cells in 33% of the T cell clones, and 1 out of 94 clones expressed the transposase but without remobilization of the transposon [71]. Even though they did not observe any negative effects on T cells, other investigators have reported that continuous expression of SB100X and SB100Xo (a codon-optimized variant) can lead to cell stress, premitotic arrest, and apoptosis, which was reduced when using the transposase delivered as mRNA [73,74]. Thus, the use of mRNA to encode the transposase is advantageous because it decreases the risk of long-term presence in genetically modified cells. The only transiently needed transposase mRNA degrades without genomic integration. Additionally, following delivery, the transposase translocates to the cytoplasm, where it is directly available for translation [75].

Here, we have examined the use of the SB100X transposase delivered as mRNA to transfect RPE cells with the Venus reporter gene and the PEDF transgene to increase the biosafety profile of our gene therapeutic approach to treat nvAMD. We showed the successful transfection and long-term expression in ARPE-19 and primary human RPE cells by delivering SB100X as mRNA. The results were comparable to those where SB100X-DNA was used.

Our results showed that the use of SB100X-mRNA is a promising tool for the engineering of a gene therapy medicinal product for the treatment of nvAMD using autologous cells secreting rPEDF. Further improvement of transfection efficiency, stability, and reproducibility in primary human RPE cells is desired before the system can be transferred to the treatment of other diseases where long-term expression of a therapeutic product is required.

## 2. Materials and Methods

### 2.1. Plasmids

The construction of the plasmid encoding the PEDF gene (pT2-CMV-PEDF/EGFP) has been described previously [53]. The PEDF and EGFP (enhanced green fluorescent protein) sequences are linked via an internal ribosome entry site (IRES) sequence to enable co-translation of two separate proteins. The plasmids pT2-CAGGS-Venus, pCMV(CAT)T7-SB100X (pSB100X), pCMV(CAT)T7-SB(-) (pSB-), and pcGlobin2-SB100X were constructed by Lajos Mátés and kindly provided by Prof. Zsuzsanna Izsvák (Max-Delbrück Center, Berlin, Germany) [51].

### 2.2. In Vitro mRNA Transcription

The in vitro transcription of SB100X-mRNA was performed using the pcGlobin2-SB100X plasmid as template. The mMessage mMachine^®^ T7 kit (Ambion, Austin, TX, USA) was used according to the manufacturer’s instructions. Briefly, the template plasmid was linearized using the restriction enzyme XbaI (New England Biolabs, Ipswich, MA, USA) followed by proteinase K (Ambion) treatment; after the transcription reaction was completed, plasmid DNA was removed using Turbo DNase (Ambion), and the mRNA was purified using the MEGAclear Kit (Ambion). Aliquots were stored at −80 °C until use. The success of the mRNA transcription was analyzed by gel electrophoresis using a 1% Tris-Borate-EDTA (TBE) agarose gel to determine RNA fragment size.

### 2.3. Cultivation of ARPE-19 Cells

ARPE-19 cells (ATCC No. CRL-2302) [76] were maintained in Dulbecco’s modified Eagle’s medium (DMEM)/Ham’s F-12 (Biochrom AG, Berlin, Germany) supplemented with 10% fetal bovine serum (FBS; PAA Laboratories, Colbe, Germany), 80 U/mL penicillin and streptomycin (Lonza, Basel, Switzerland), and 2.5 μg/mL amphotericin B (PAA Laboratories) at 37 °C in a humidified atmosphere of 95% and 5% CO_2_. Medium was changed three times per week. Cells were subcultivated weekly at a ratio of 1:10. Trypsinization of cells was performed using 0.05% Trypsin/0.02% EDTA (PAA Laboratories) for 7 min at 37 °C in a humidified atmosphere. The enzymatic reaction was stopped by adding the same volume of complete culture medium containing 10% FBS.

### 2.4. Isolation and Cultivation of Primary Human RPE Cells

Human donor eyes were obtained from the Aachen Cornea Bank (Department of Ophthalmology, University Hospital RWTH Aachen, Germany). The eyes were collected after informed consent was obtained in accord with the declaration of Helsinki. All procedures for the collection and use of human samples were approved by the institutional ethics committee.

Primary RPE cells from 12 different donors (mean age 65.7 ± 9.8 years; 7 males and 5 females, post-mortem time 37.5 ± 16.6 h) were isolated and cultured for 26.4 ± 11.3 days before being used for transfection experiments. Detailed donor characteristics can be found in Appendix A in the Appendix A. The isolation of primary human RPE cells was performed as described previously [54,77]. Briefly, globes were transected 3 mm below the limbus by the Aachen Cornea Bank, the cornea was taken, and the remaining tissue could be used for cell isolation. To isolate RPE cells, iris, lens, vitreous, and retina were removed, and the eye cup was filled with 1 mL DMEM/Ham’s F-12 (Biochrom AG) supplemented with 10% fetal bovine serum (FBS; PAA Laboratories), 80 U/mL penicillin and streptomycin (Lonza), and 2.5 μg/mL amphotericin B (PAA Laboratories). RPE cells were dislodged by brushing the surface with a fire-polished glass spatula. Cells from one eye were apportioned into 3 wells of a 24-well culture plate. Cultures were maintained at 37 °C, 5% CO_2_ in a humidified atmosphere. Medium was changed twice per week until cultures reached confluence and were used for transfection experiments.

### 2.5. Electroporation and Cultivation of RPE Cells

Electroporation of ARPE-19 and primary human RPE cells was performed as described previously using the Neon^®^ Transfection System (Thermo Fisher Scientific, Waltham, MA, USA) and the SB100X transposon system [53,54,77]. Then, 10^5^ cells in 11 μL resuspension buffer R were mixed with 2 μL of purified plasmid/mRNA mixture (total concentration 500 ng) containing purified plasmid encoding the Venus gene or the PEDF-EGFP genes (transposon) and the SB100X transposase delivered as DNA or mRNA. The optimal ratio of transposase to transposon was determined by varying the ratios of SB100X-mRNA transposase to transposon plasmid (the exact amounts of used plasmids for each ratio in ng and pmol are available from Appendix A in the Appendix A). Transfected ARPE-19 cells were seeded in 6-well plates at a density of 10,400 cells/cm^2^, and transfected primary hRPE cells were seeded in 24-well plates at a density of 50,000 cells/cm^2^ in culture medium with 10% FBS without antibiotics and antimycotic to reduce cell stress and damage. Penicillin, streptomycin, and amphotericin B were added with the first medium exchange 3 days after electroporation. Medium of ARPE-19 cells was changed 3 times per week; the cells were subcultivated once per week, fluorescence was measured regularly over the entire time of the experiment. For human primary cells, medium was changed twice per week with medium supplemented with 10% FBS; at confluence, FBS was reduced to 1% to reduce proliferation until the cultures were terminated. Transfection efficiency in primary cells transfected with the Venus gene was determined at confluence. PEDF-transfected cells were maintained in culture and the media used to determine PEDF secretion by Western blot analysis.

### 2.6. Fluorescence Imaging

Brightfield and fluorescence imaging were performed on confluently grown cultures using the microscope DM6000B (Leica, Wetzlar, Germany). Exposure time was set the same for all cultures and timepoints of the same experiment to allow direct comparison of the fluorescence intensity.

### 2.7. Flow Cytometric Analysis

To determine transfection efficiency, aliquots of trypsinized ARPE-19 cells transfected with the pT2-CAGGS-Venus and the pT2-CMV-PEDF/EGFP constructs and primary hRPE cells transfected with the pT2-CAGGS-Venus construct were washed 3 times with PBS, resuspended in 100 µL PBS, and analyzed for fluorescence using the FACSCalibur instrument (Becton Dickinson). The mean fluorescence intensity (MFI) was calculated from all measured cells in relative fluorescence units (RFU).

### 2.8. Protein Purification, SDS-PAGE, and Western Blot Analysis

Purification of His-tagged proteins from culture supernatants was performed using Ni-NTA metal-affinity chromatography (Qiagen, Hilden, Germany) according to the manufacturer’s instructions and as detailed previously [53]. Purified proteins were separated on a 10% SDS-PAGE and transferred to a 0.45 µm pore-size nitrocellulose membrane. Recombinant, his-tagged proteins were detected using penta-his tag antibodies (mouse monoclonal, Qiagen, catalogue number 34660, public identifier: AB_2619735, diluted 1:500 in 3% bovine serum albumin (BSA)/Tris-buffered saline (TBS)) and a secondary antibody conjugated with horse-radish peroxidase (HRP) (Polyclonal Rabbit Anti-Mouse HRP, Agilent, Dako, catalogue number P0260, public identifier: AB_2636929, diluted 1:1000 in 10% milk powder/TBS). Protein bands were visualized by chemiluminescence using the LAS-3000 imaging system (FujiFilm, Tokyo, Japan) and evaluated using the image processing program ImageJ (Rasband, W.S., ImageJ, U.S. National Institutes of Health, New York, NY, USA).

### 2.9. Polymerase Chain Reaction (PCR) and Real-Time Quantitative PCR

Total RNA isolation was performed using the RNeasy Mini kit, QIAShredder kit, and RNase-free DNase Set (Qiagen). Reverse transcription was carried out using the Reverse Transcription system (Promega, Madison, WI, USA). Genomic DNA isolation was performed using the QIAamp DNA Mini Kit (Qiagen). All kits were used according to the manufacturer’s recommendations. PCR was performed using the Go Taq^®^ Hot Start Polymerase (Promega). The primers listed in Table 1 were employed at a concentration of 10 pmol/µL. cDNA was diluted to a concentration corresponding to 2 ng of initially used RNA per PCR reaction, and genomic DNA (gDNA) was diluted to a concentration of 5 ng/µL. The amount of 5 µL of diluted DNA was used for the PCR reaction that was carried out as follows: initial denaturation at 94 °C for 2 min, followed by 40 cycles with denaturation at 94 °C for 30 s, annealing at 56 °C or 58 °C for 30 s, and elongation at 72 °C for 30 s. After a final elongation step at 72 °C for 5 min, the products were separated by agarose gel electrophoresis. Products with a size > 100 bp were separated on 2% agarose gels, and smaller fragments were separated on 3% gels.

Real-time quantitative (q)PCR was performed using the LightCycler 1.2 Instrument (Roche Diagnostics, Basel, Switzerland) and the LightCycler FastStart DNA Master SYBR Green I kit (Roche Diagnostics) according to the manufacturer’s instructions. The cDNA samples were run in duplicate. The following thermal cycler conditions were used: initial denaturation at 95 °C for 10 s, 60 cycles with denaturation at 95 °C for 10 s, annealing at 60 °C for 8 s, and elongation at 72 °C for 15 s, followed by a melting curve to confirm the specific amplification from each primer pair even at high Ct values due to low expression levels. Data were analyzed using the LightCycler Software 3.5.3 (Roche Diagnostics) and the comparative CT method for relative gene expression [78].

### 2.10. Statistical Analysis

Statistical analysis was performed using the test indicated in the figure legend and GraphPad Prism software version 8 (GraphPad, San Diego, CA, USA).

## 3. Results

### 3.1. Transfection Efficiency and Long-Term Venus Expression in ARPE-19 Cells Using the SB100X Transposase as mRNA

To determine optimal transfection efficiency, different ratios of SB100X-mRNA and pT2-CAGGS-Venus of 1:1 to 1:36 were assessed and observed over a period of up to 583 days post-transfection (Figure 2). SB100X-DNA (pSB100X) as positive control and inactive transposase (pSB-) as negative control were used in a ratio of 1:28. Initial transfection efficiency in Venus-transfected cells one week post-transfection was 98–100% in SB100X-DNA transfected cultures, and in all cultures, transfection was carried out with SB100X-mRNA, except for the ratio 1:1 (84.7%) and the pSB- control (93.7%), with MFI ranging from 1242 RFU (ratio 1:1) to 4398 RFU (SB100X-DNA) (Figure 2A). Long-term expression was observed in four out of ten cultures of mRNA-transfected ARPE-19 cultures (Figure 2B), two cultures were monitored for 583 days (ratio 1:1 and 1:16) and two cultures for up to 270 days (ratio 1:28 and 1:36). The percentage of Venus-positive cells in the culture transfected with SB100X-DNA fell below 10% at 81 days, as also observed by fluorescence microscopy (Figure 2C); the culture was terminated at 102 days, when the percentage was below 1%. In all other cultures, Venus expression decreased below 10% during the 160 days post-transfection. Fluorescence micrographs of the four long-term cultures transfected with pT2-CAGGS-Venus showed that the cultures progressed from cultures of mixed fluorescence intensity to cultures with similar fluorescence intensity.

Using SB100X-mRNA and the pT2-CAGGS-Venus transposon at a ratio of 1:16 resulted in a high initial transfection efficiency and stable and long-term expression of the transgene without an appreciable reduction in the number of Venus-positive cells during the first weeks after transfection. Therefore, this ratio was used for further experiments.

### 3.2. Comparison of SB100X-DNA and -mRNA for the Transfection of ARPE-19 Cells and Analysis of SB100X Expression

After the initial titration experiment, a comparison of transfection efficiency and stability of transgene expression was examined by transfecting in triplicate ARPE-19 cells with the Venus gene and SB100X-DNA at a transposase:transposon ratio of 1:28 or SB100X-mRNA at a ratio of 1:16. Transfected cells were cultured for 182 days and monitored weekly by fluorescence microscopy and flow cytometric analysis. SB100X expression and genomic analysis were performed over the first 28 days post-transfection.

The flow cytometric analysis showed an initial transfection efficiency of 100% in all samples at 3 days post-transfection (Figure 3A), with an MFI between 5501 RFU (DNA A) and 6713 RFU (DNA C) (Figure 3B).

At 35 and 42 days after transfection, two cultures which were transfected with SB100X-DNA (samples DNA A and DNA B) showed reduction in the number of fluorescent cells, which fell below 1% at days 112 and 126 (Figure 3A). Both cultures were terminated at day 126. The MFI was reduced simultaneously (Figure 3B). In the third culture transfected with SB100X-DNA (sample DNA C), a reduction in fluorescent cells began at 105 days of culture and fell below 20% at day 133 and below 1% at day 182, when this experiment was ended. In all three cultures transfected with SB100X-mRNA, the number of fluorescent cells remained constant at 100% over the entire observation period of 182 days (Figure 3A). The MFI in these samples increased during the first 70 days and stayed stable at approximately 9000 RFU from day 70 to day 182 (Figure 3B). Fluorescence microscopy micrographs (Figure 3C) showed a similar pattern: reduced fluorescence by ARPE-19 cells transfected with SB100X-DNA but not by cells transfected with SB100X-mRNA. Brightfield micrographs are displayed exemplarily for day 7, showing confluent growth and healthy morphology of the cells, as observed at each timepoint.

In addition to the fluorescence analysis, the SB100X sequence was detected in genomic DNA and total RNA by PCR (Figure 3D). The genomic sequence was detected in the three DNA-transfected samples 3, 7, and 14 days after transfection but not in the SB100X-mRNA transfected cells, as expected. Additionally, expression of the transposase was detected in the SB100X-DNA transfected samples 3 and 7 days after transfection but not in the SB100X-mRNA transfected samples, indicating lower mRNA levels in mRNA transfected cells compared with DNA transfected cells. After 21 and 28 days, no transposase was detected for either mRNA or DNA samples.

### 3.3. Transfection Efficiency and Long-Term PEDF Secretion in ARPE-19 Cells Using the SB100X Transposase as mRNA

ARPE-19 cells were transfected with SB100X-mRNA and pT2-CMV-PEDF/EGFP using different ratios of 1:1 to 1:36 and observed over a period of up to 500 days post-transfection. SB100X-DNA (pSB100X) as positive control and inactive transposase (pSB-) as negative control were used in a ratio of 1:28.

Cells transfected with SB100X-mRNA showed initial transfection efficiencies ranging from 8% at a 1:1 ratio of SB100X-mRNA to transposon to 42% at a ratio of 1:36 one week post-transfection (Figure 4A, bars, left axis). The highest MFI was detected in cells transfected with SB100X-mRNA at a ratio of 1:24 and 1:36 (MFI of 105 RFU), the lowest MFI in cells transfected with SB100X-mRNA at a ratio 1:1 (MFI of 7 RFU) (Figure 4A, dots, right axis). Figure 4B illustrates the number of EGFP-positive ARPE-19 cells transfected with the construct pT2-CMV-PEDF/EGFP that were followed in culture for 500 days for the first experiment (ratio 1:1–1:20) and 277 days for the second experiment (ratio 1:16–1:36), respectively. Over time, an enrichment of EGFP-positive cells was observed, which resulted in cultures of greater than 90% EGFP-positive cells at different time points (SB100X DNA at day 87, mRNA 1:20 at day 249, mRNA 1:24 at day 277, and mRNA 1:36 at day 144). rPEDF secreted into the media was detected by Western blot in all cultures one week post-transfection; note that even ARPE-19 cells transfected with the inactive transposase showed secreted rPEDF at 7 and 8 days after transfection, but 13 days after transfection, PEDF could be no longer detected (second series, bottom) (Figure 4C).

Using the SB100X-mRNA in combination with the pT2-CMV-PEDF/EGFP transposon plasmid, cells transfected at a ratio of 1:20 showed the highest PEDF-secretion after 277 days. Since PEDF-secretion is the most important factor, this ratio was used for further studies in ARPE-19 cells.

### 3.4. Gene Expression Analysis of Transfected ARPE-19 Cells

One year after transfection, mRNA expression levels of different RPE markers (Figure 5A) as well as VEGF, endogenous PEDF, rPEDF, EGFP/Venus, and SB100X were analyzed (Figure 5B). Genomic DNA analysis was performed to verify sequences that are involved in the transfection: the SB100X transposase, the respective transposon via the IRDRL sequence, EGFP, and Venus as well as recombinant PEDF (Figure 5C). Analysis of the housekeeping gene GAPDH and the single-copy gene RPPH1 demonstrated similar cDNA-synthesis for RT-PCR and a similar amount of gDNA for PCR in all samples. For these analyses, cells from the transfection experiments shown in Figure 1 and Figure 4 were used, which still showed transgene expression 1 year post-transfection: pT2-CMV-PEDF/EGFP + DNA, pT2-CAGGS-Venus + mRNA 1:1 and 1:16, in comparison with non-transfected control cells.

The transfection of the ARPE-19 cells with pT2-CAGGS-Venus or pT2-CMV-PEDF/EGFP did not influence the expression of the specific RPE cell markers CTSD, KRT8, or ZO-1 or the expression of endogenous PEDF, VEGF, or VEGF-R2 (Figure 5A,B). In SB100X-DNA-transfected cells, the expression of CRALBP was slightly decreased compared with the non-transfected controls and the other transfected cultures. Expression of the PEDF and Venus transgenes was observed in the respective transfected cultures, whereas expression of the SB100X transposase, either delivered as mRNA or DNA, was not detected in any of the transfected cultures. In addition, in the genomic DNA samples, the SB100X sequence was not detected, whereas the transgenes PEDF, EGFP, and Venus were detected in the genomic DNA of the respective cultures, indicating the insertion of the transposon into the genome (Figure 5C). However, the detection of the transposon by the IRDRL sequence was only positive in the PEDF/EGFP-transfected cells but not in the Venus-transfected cells.

Expression of the genes CRALBP, CTSD, KRT8, ZO-1, VEGF, endogenous PEDF, and total PEDF (endogenous PEDF plus recombinant PEDF) was also analyzed by qPCR 1 year post-transfection in the following samples: pT2-CMV-PEDF/EGFP + DNA, pT2-CMV-PEDF/EGFP + mRNA 1:20, 1:24, and 1:36, pT2-CAGGS-Venus + mRNA 1:1 and 1:16 (Figure 5D–F). The relative gene expression in transfected cells was calculated and normalized to the relative gene expression in non-transfected control cells, and the multiple *t*-test was used to analyze the differences in transfected cells compared with non-transfected control cells. For none of the studied endogenous genes was gene expression changed more than 10-fold with *p* ≥ 0.05 (Figure 5D,E). The expression of CRALBP was slightly increased in the cultures transfected with pT2-CMV-PEDF/EGFP and SB100X-mRNA at ratios of 1:20 and 1:36. Expression of total PEDF increased in PEDF-transfected cells; in the cells transfected with SB100X-DNA, the relative PEDF gene expression was 10^8^-fold higher than in non-transfected control cells, and in cells transfected with SB100X-mRNA, relative PEDF gene expression showed a 10^4^-fold increase at a ratio of 1:24 and a 10^5^-fold increase at a ratio of 1:36 and 1:20 transposase to transposon compared with non-transfected control cells (Figure 5F).

### 3.5. Transfection of Primary Human RPE Cells

Primary human RPE cells were transfected with the pT2-CAGGS-Venus and SB100X-mRNA at ratios ranging from 1:12 to 1:24. This smaller range of ratios (1:12–1:24 instead of 1:1–1:36) was chosen since less human primary cells per donor were available, and because in ARPE-19 cells, the highest transfection efficiencies were observed within this range. SB100X-DNA and inactive transposase were used as respective controls at a ratio of 1:16 [41]. Cells from six donors were transfected and followed separately. Detailed donor characteristics are listed in Appendix A (Appendix A).

Transfected cells were photographed in bright field and under fluorescence light and analyzed by flow cytometry at 10–14 days after transfection. In all experiments, the number of fluorescent cells and MFI were similar for cells transfected with SB100X-mRNA and SB100X-DNA; cells transfected with SB- showed a weaker fluorescence (Figure 6A). The transfection efficiency and the MFI were highly variable among the six donors, ranging from more than 80% in cells from Donor 1 to less than 5% in cells from Donor 6. Fluorescence microscopy of transfected RPE cells from Donor 1 and Donor 6 is illustrated in Figure 6B, which underlines the high variability of the fluorescence measurements.

Primary hRPE cells isolated from 10 donors were transfected with SB100X-mRNA and pT2-CMV-PEDF/EGFP at ratios between 1:8 and 1:28 transposase to transposon. Note that not all ratios could be tested in all donors depending on the number of cells available. Additionally, cells were transfected with pT2-CMV-PEDF/EGFP and SB100X-DNA or SB- at a transposase:transposon ratio of 1:20. The donor characteristics are summarized in Appendix A (Appendix A). Purified culture media were analyzed by Western blot for rPEDF secretion at 1 week and then every 4–6 weeks for up to 383 days. Relative signal intensities were determined using ImageJ software, data were normalized to the SB100X-DNA control from the same donor at the same time point and electrophoresed on the same gel. The results are summarized in five time periods from 5.4 ± 1.96 to 371.3 ± 10.87 days (Figure 7A). Representative Western blot images of Donor 5 are shown in Figure 7B.

His-tagged rPEDF was detected during the first week after transfection in all donors (period 1), and the average PEDF secretion was higher or similar in all cells transfected with SB100X-mRNA compared with SB100X-DNA transfected cells. By 371 ± 10.87 days, cells from the majority of donors no longer expressed PEDF, and those that did, secreted lower levels compared with cells transfected using SB100X-DNA. Transfection of cells with pT2-CMV-PEDF/EGFP and pSB- resulted in transient secretion of rPEDF 5 days after transfection, and rPEDF was no longer detected after 26 days. For SB100X-DNA and -RNA transfected cultures, rPEDF secretion was high after 5 and 26 days and was reduced but still detectable after 323 days.

## 4. Discussion

Transposons, such as the hyperactive *Sleeping Beauty* (SB100X) transposon system, integrate non-viral vectors, exhibiting several advantageous features with a high safety profile and efficiency [51,79,80]. The *Sleeping Beauty* transposon system is already being used overall in 14 clinical trials in the United States and Europe, mostly to modify ex vivo T cells followed by implantation into leukemia patients [67,81,82,83]. The transposon system is also applied in the establishment of cell lines to produce recombinant therapeutic proteins with high and stable expression levels [72,84].

To avoid the risk of random integration of the transposase sequence, remobilization of the transposon, and potential cytotoxic or genotoxic effects caused by continued SB100X expression, we have investigated the use of SB100X-mRNA as source of transposase to transfect retinal pigment epithelial cells.

Here, we have shown that mRNA encoding SB100X can be efficiently delivered to RPE cells by electroporation, as evidenced by stable integration of the transposon into the host cells’ genome. High transfection efficiency with *Sleeping Beauty* requires that the ratio of transposase to transposon is optimized to prevent overproduction inhibition (OPI) [61,72,85] and to optimize transfection efficiency, which highly varies depending on the cell line, cell type, and donor species [86]. Previous studies have shown 100% transfection efficiency of ARPE-19 cells with the Venus transgene and SB100X as DNA with a transposase to transposon ratio of 1:4 to 1:40 [53]. Since these experiments have been performed already using the same transposon plasmids in combination with SB100X delivered as DNA, we renounced the performance of replicates for each ratio and focused on the long-term expression of the transgenes in these experiments. In four out of ten cultures, long-term expression of Venus could be observed in ARPE-19 cells transfected with pT2-CAGGS-Venus and SB100X-mRNA at different ratios. After 583 and 270 days, cultures transfected with SB100X-mRNA at ratios of 1:1, 1:16, 1:28, and 1:36 consisted of 100% fluorescent cells, but with different MFI ranging from 2599 RFU (mRNA 1:28) to 9793 RFU (mRNA 1:36), respectively, indicating that the cells contain different copy numbers of the transgene. These results imply that the ratio of 1:16 should be used for future experiments since it led to a high initial transfection efficiency and stable and long-term expression of the transgene without reduction in the number of Venus-positive cells during the first weeks after transfection.

The transfection of ARPE-19 cells with pT2-CMV-PEDF/EGFP showed transfection efficiencies between 8% and 42% together with high rPEDF secretion rates. In this construct, the EGFP expression cassette is linked by an IRES-sequence behind the PEDF sequence, resulting in a lower translation of the EGFP sequence, which could explain the lower percentage of fluorescent cells compared with Venus-transfected cells. Using non-viral gene transfer methods, cargo capacity is high, but the transfection rate is inversely correlated with plasmid size [87,88,89]. The pT2-CMV-PEDF/EGFP plasmid with a size of 7259 bp is 18.6% larger than the pT2-CAGGS-Venus construct with 6123 bp, which could explain the results. For the gene therapeutic treatment of AMD patients, we plan to use a smaller pFAR4-based plasmid construct, encoding only for PEDF, which will increase gene transfer rates [55]. Three out of ten cultures and the SB100X-DNA control culture showed an increase in the number of fluorescent cells over time and long-term transgene expression. Cultures transfected with the inactive transposase (pSB-) and pT2-CAGGS-Venus or pT2-CMV-PEDF/EGFP showed high initial fluorescence rates that decreased during the first two weeks, indicating that in the presence of SB100X as mRNA or DNA, the transgenes had become integrated into the host cells’ genome, which is necessary for long-term expression of the transgene. 

The gene expression analysis of RPE and epithelial cell markers in the transfected ARPE-19 cells revealed a gene expression profile of analyzed markers roughly similar to non-transfected control cells. Only for CRALBP, which plays a role in the visual cycle, expression was slightly changed in some PEDF-cultures, which could be an indication that the increased PEDF production affects the expression of CRALBP. The decrease in fluorescence or PEDF secretion in some cultures of ARPE-19 cells transfected with the Venus or PEDF gene may be the result of a difference in growth rates depending on the number of transgene copies integrated, as discussed in detail below.

In the follow-up experiment, ARPE-19 cells were transfected in triplicate using either SB100X-mRNA or -DNA. The three mRNA-transfected cultures showed stable expression of the Venus transgene, whereas the SB100X-DNA transfected cultures decreased in the number of fluorescent cells. PCR analysis showed the presence of SB100X-DNA and expression in the three cultures transfected with SB100X-DNA during the first 14 days after transfection. Different explanations are possible for the decrease of fluorescence in the SB100X-DNA transfected cultures: it might be possible that the transposase sequence integrated into the genome, resulting in remobilization of the transgene and loss of the gene or death of these cells, but it was not detectable by PCR. However, this would not explain the decrease in the different SB100X-mRNA transfected cultures in the first experiment (Figure 2). Probably, the non-transfected cells have a benefit in proliferation, and after repeated sub-cultivation, they may overgrow the transfected cells. Although the initial transfection efficiency measured was 100% with the pT2-CAGGS-Venus plasmid, it could have happened that some cells were not stably transfected. The cultures which stayed at 100% changed into homogeneous cultures after several weeks, indicating that the cells with a specific copy number proliferate faster than the other cells. This could also explain the enhancement of the pT2-CMV-PEDF/EGFP transfected cells; it is possible that in these cultures, the PEDF-transfected cells had a benefit in proliferation and overgrew the non-transfected cells, as already shown by Johnen et al. [90]. As this effect is only possible in cell lines which have a high proliferation rate and are sub-cultivated weekly, it should not be seen in primary cells which maintain in the culture dish after transfection without sub-cultivation.

Primary RPE cells isolated from six donors were transfected with the Venus gene and SB100X-mRNA or SB100X-DNA. Transfection efficiencies ranged from 5% to 80% between cells from the different donors; similar efficiencies were obtained whether SB100X-mRNA or -DNA was used in cells isolated from the same donor. However, efficiency was significantly lower using inactive transposase. The difference in the transfection efficiency is not correlated to the age of the donor or the length of time that the cells were in culture before transfection. Here, we show that using SB100X as mRNA, transfection efficiency similar to SB100X-DNA transfected cells can be obtained with a ratio of transposase to transposon between 1:12 and 1:24. Long-term continuous and stable secretion was evidenced by analyzing rPEDF secretion of PEDF-transfected primary human RPE cells isolated from 10 different donors using either SB100X-DNA or mRNA. These results are comparable to the results of Bire et al. who used the *piggyBac* transposon system to transfect HeLa cells and showed that the use of transposase mRNA leads to a comparable transfection efficiency with transposase DNA [91]. The investigators also showed that the expression period of the transposase in mRNA transfected cells was 36 h compared with 48 h for cells transfected with the DNA transposase. Wilber et al. also showed the successful transfection and transposition in vitro and in vivo using the less active form *Sleeping Beauty* 11 mRNA as the source of transposase [69,70].

We have established an efficient and safe transfection method to transfect pigment epithelial cells using the *Sleeping Beauty* transposon system with SB100X delivered as mRNA. These results show that the use of SB100X-mRNA is a promising tool for the engineering of a gene therapeutic medicinal product for the treatment of nvAMD using autologous cells secreting rPEDF. The method has a high potential to be transferred to other cell types and gene therapeutic treatment strategies where high and long-term gene expression is desired. Further investigations will focus on the high variation in primary human cells and the question of why some cultures are stably transfected while others show decreased transgene expression over time, including the kinetics of SB100X expression and transposition, followed by the analysis of the biosafety, the long-term expression, and the efficiency of the treatment in in vivo models.

## Figures and Tables

**Figure 1 biomolecules-13-00658-f001:**
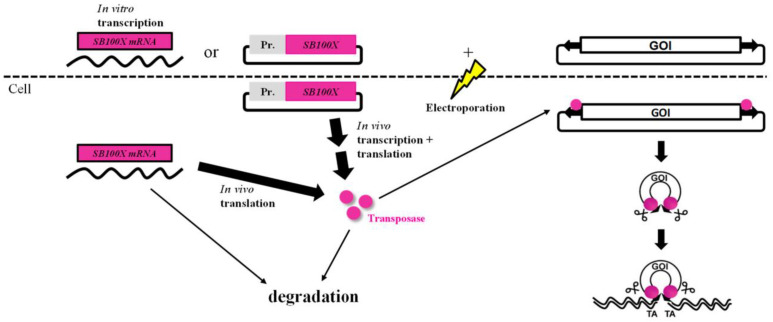
Non-viral Sleeping Beauty Transposon System. The transposon plasmid encoding the gene of interest and the SB100X transposase, delivered as DNA or mRNA, are introduced into the cells via electroporation. The transposase binds the TIRs flanking the GOI, cuts it out, and inserts it into the host cells’ genome to allow for stable and long-term expression and secretion of the therapeutic protein, such as PEDF.

**Figure 2 biomolecules-13-00658-f002:**
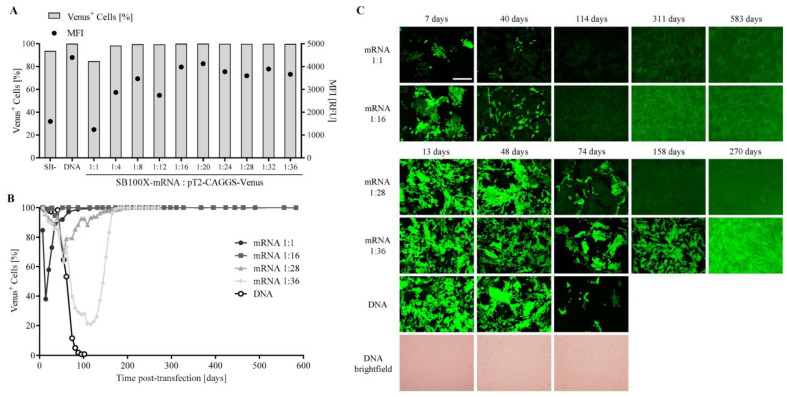
Analysis of transfection efficiency and long-term expression in ARPE-19 cells transfected with pT2-CAGGS-Venus. ARPE-19 cells were transfected with pT2-CAGGS-Venus and SB100X-mRNA at ratios from 1:1 to 1:36 with a total amount of 500 ng nucleic acid. SB100X-DNA served as the positive control, the inactive transposase pSB- as negative control, both at a ratio of 1:28. Cultures were observed regularly and were terminated when the percentage of Venus-positive cells decreased to less than 10%. (**A**) Initial transfection efficiency was determined 1 week after transfection by flow cytometric analysis, measuring the percentage of Venus-positive cells (bars, left axis) and the MFI (dots, right axis). (**B**) Two cell cultures transfected with SB100X-mRNA were monitored for 583 days (ratio of 1:1 and 1:16) and two cultures up to 270 days (ratio 1:28 and 1:36) by flow cytometric analysis, shown is the percentage of Venus-positive cells. (**C**) Fluorescence micrographs of the four long-term cultures show the progression of the cultures from cultures of mixed fluorescence intensity to cultures with similar fluorescence intensity. Note that for the DNA control culture, a reduction in fluorescence was observed. Brightfield micrographs are shown exemplarily for the DNA control culture, indicating the confluent growth of the cultures. Scale bar: 500 µm. The fluorescence analysis of all cultures over the entire observation period can be found in the Appendix A (Appendix A).

**Figure 3 biomolecules-13-00658-f003:**
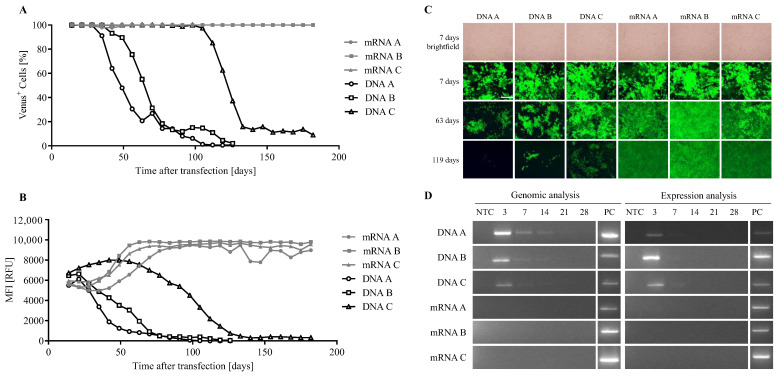
Expression analysis of ARPE-19 cells transfected with the pT2-CAGGS-Venus transposon plasmid together with SB100X-DNA or -mRNA. ARPE-19 cells were transfected with the pT2-CAGGS-Venus construct using SB100X-mRNA (ratio 1:16) or -DNA (ratio 1:28) and analyzed weekly by flow cytometry (**A**,**B**) and fluorescence microscopy ((**C**), scale bar: 500 µm) for at least 126 days. (**D**) Genomic DNA and RNA of the transfected cells were isolated at 3, 7, 14, 21, and 28 days and analyzed by PCR to detect SB100X. Samples from ARPE-19 cells transfected with pT2-CAGGS-Venus and SB100X-DNA 3 days after transfection were used as positive control (PC) for the detection of the SB100X sequence.

**Figure 4 biomolecules-13-00658-f004:**
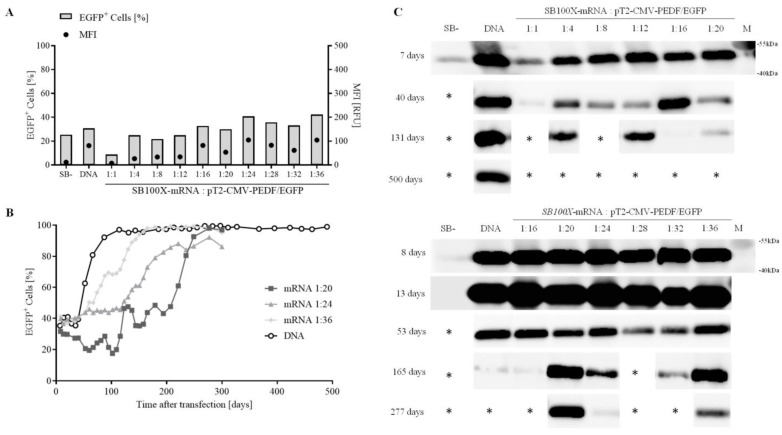
Analysis of transfection efficiency and long-term expression in ARPE-19 cells transfected with pT2-CMV-PEDF/EGFP. ARPE-19 cells were transfected with pT2-CMV-PEDF/EGFP and SB100X-mRNA at ratios from 1:1 to 1:36 with a total amount of 500 ng nucleic acid. SB100X-DNA served as the positive control, the inactive transposase pSB- as negative control, both at a ratio of 1:28. Cultures were observed regularly and were terminated when no rPEDF signal was detected by Western blot. (**A**) Initial transfection efficiency was determined 1 week after transfection by flow cytometric analysis, measuring the percentage of EGFP-positive cells (bars, left axis) and the MFI (dots, right axis). (**B**) Three cultures (ratio of 1:20, 1:24, and 1:36) were monitored over a period of 277 days and the SB100X-DNA control culture for 500 days. (**C**) Western blot analysis was carried out on His-tag purified proteins from culture media of transfected cells using penta-his tag antibodies at different time points for 2 experimental series (top and bottom). Cultures were terminated when only a low or no rPEDF secretion was detectable by Western Blot or when the percentage of fluorescent cells decreased below 10%. Terminated cultures are labeled with *. Please note that not all performed Western Blots are shown here. All Western Blots can be found in the Appendix A (Appendix A) together with the fluorescence analysis of all samples for the entire duration of the experiments.

**Figure 5 biomolecules-13-00658-f005:**
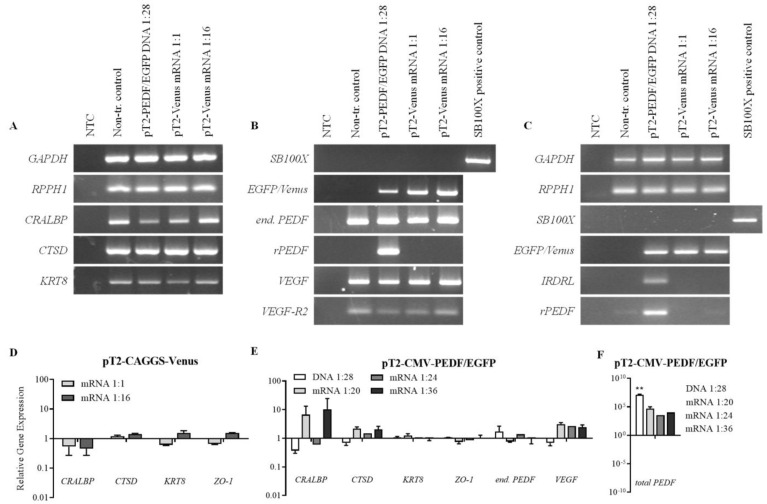
Gene expression and genomic analysis of ARPE-19 cells transfected with pT2-CAGGS-Venus or pT2-CMV-PEDF/EGFP compared with non-transfected control cells one year after transfection. One year after transfection, RNA and DNA were isolated from transfected and non-transfected ARPE-19 cell cultures, followed by PCR and agarose-gel electrophoresis (**A**–**C**) or real-time RT-qPCR (**D**–**F**). Panel A illustrates the mRNA expression levels of several endogenous genes in transfected and non-transfected cells. Panel B shows the expression of transfection related as well as PEDF and VEGF expression. Panel C illustrates the genomic DNA analysis of various sequences involved in the transfection. (**D**,**E**) Relative gene expression of different endogenous genes was analyzed by RT-qPCR in pT2-CAGGS-Venus or pT2-CMV-PEDF/EGFP transfected cells compared with non-transfected cells. (**F**) In PEDF-transfected cells, the expression of total PEDF was also analyzed. Gene expression changes were calculated using the ΔΔCt method, GADPH was used as housekeeping gene, and the results were expressed relative to the control group (non-transfected cells). All bars show mean ± SD from the two technical replicates. NTC—no template control, Non-tr. Control—non-transfected control. The multiple unpaired t-test was used to analyze the differences in transfected cells compared with non-transfected control cells (all samples *p* ≥ 0.05, except for ** *p* = 0.0046).

**Figure 6 biomolecules-13-00658-f006:**
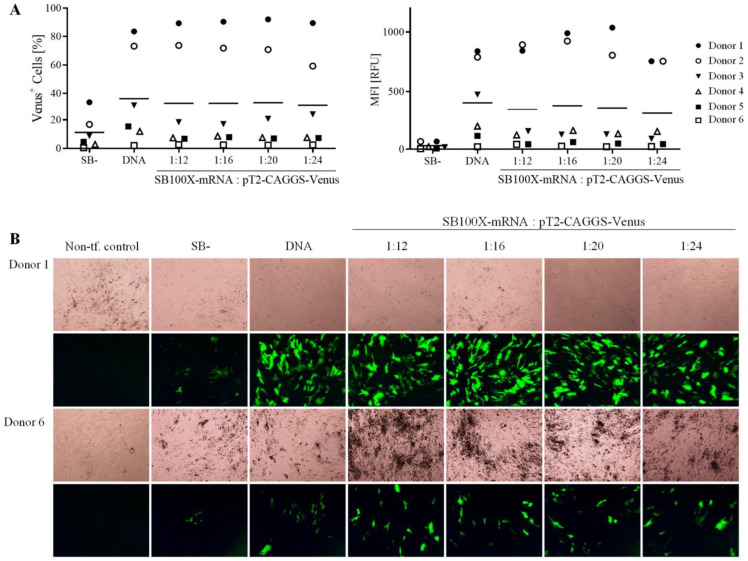
Analysis of primary human RPE cells transfected with pT2-CAGGS-Venus and SB100X delivered as mRNA using different transposon:transposase ratios. hRPE cells from 6 different donors (donor age 68.2 ± 7.9 years, 4 males and 2 females, post-mortem time 40.0 ± 22.9 h, cultivation time between isolation and transfection 24.7 ± 11.1 days) were transfected with different ratios of the pT2-CAGGS-Venus plasmid and SB100X-mRNA using electroporation. An inactive transposase was used as negative control, and SB100X-DNA was used as positive control at a ratio of 1:16. (**A**) Confluent cultures were analyzed by flow cytometry at 10–14 days after transfection. The percentage of Venus-positive cells (left graph) and the MFI (right graph) were analyzed for each individual donor (dots), and the mean for all donors was calculated (line). (**B**) Fluorescence and brightfield micrographs of confluent cultures show the variability in the number of transfected cells in Donor 1 and Donor 6, respectively (Scale bar: 500 µm).

**Figure 7 biomolecules-13-00658-f007:**
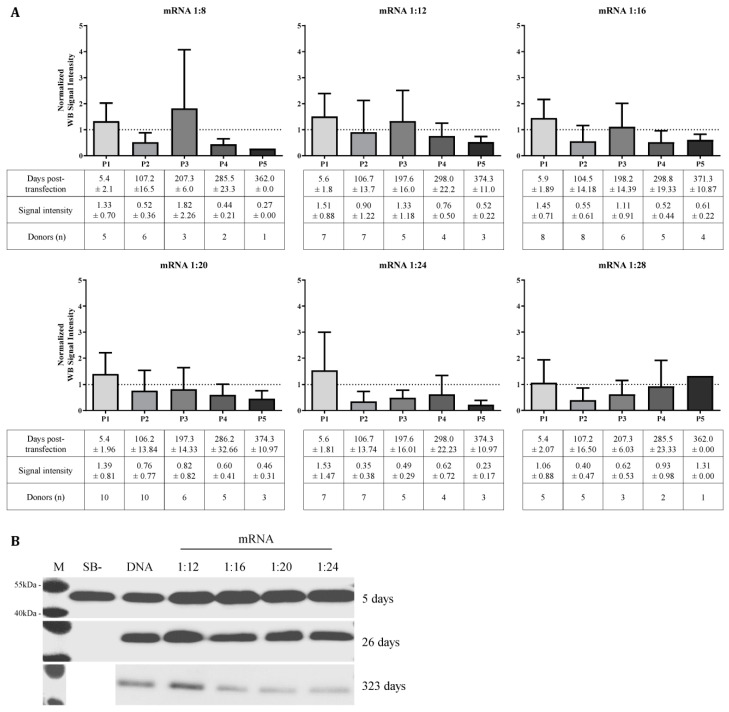
Analysis of rPEDF secretion of human primary RPE cells transfected with pT2-CMV-PEDF/EGFP and SB100X delivered as mRNA using different ratios. hRPE cells from 10 different donors (donor age 64.1 ± 10.0 years, 5 males and 5 females, post-mortem time 39.9 ± 16.3 h, cultivation time between isolation and transfection 24.0 ± 10.8 days) were transfected and kept in culture as long as PEDF secretion was detected by Western blot analysis. Supernatants were taken every 4–6 weeks, and rPEDF was detected using penta-his tag antibodies. (**A**) Signal intensity of Western blots was quantified using ImageJ software and normalized to SB100X-DNA transfected cultures (dotted line), and the results were summarized in 5 periods (P1–P5). Graph bars, days post-transfection, and signal intensity are illustrated as mean ± SD. (**B**) Representative Western blot images of Donor 5 at 5, 26, and 323 days after transfection. M—Protein marker.

**Table 1 biomolecules-13-00658-t001:** Primer sequences used for PCR and qPCR analyses, resulting fragment sizes, and annealing temperatures.

Gene	Gene ID	Sequence (5′-3′)	Fragment Size	Annealing Temperature
*GAPDH*	2597	F: AAG GTC ATC CCT GAG CTG AAC	353 nt	56 °C
		R: TTA CTC CTT GGA GGC CAT GTG		
*KRT8*	3856	F: AAG GAT GCC AAC GCC AAG TTG	360 nt	58 °C
		R: CGA TCT TCT TCA CAA CCA CGG		
*VEGF*	7422	F: AAG GAC CTA TGT CCT CAC ACC	399 nt	58 °C
		R: TAG TGA CTG TCA CCG ATC AGG		
*ZO-1*	7082	F: ACA CTG CTG AGT CCT TTG GTG	398 nt	58 °C
		R: CTA GCC AAT ACC AAC AGT CCC		
*VEGF-R2*	3791	F: ATG TGA AGC GGT CAA CAA AGT	59 nt	58 °C
		R: CTG GTC ACG TGG AAG GAG AT		
*IRDRL*	Synthetic sequence	F: CTC GTT TTT CAA CTA CTC CAC AAA TTT CT	85 nt	58 °C
		R: GTG TCA TGC ACA AAG TAG ATG TCC TA		
*rPEDF*	5176 (cDNA sequence)	F: TAC TGA GGG ACA CAG ACA CA	180 nt	56 °C
		R: AAG TCA TGC CCG CTT TTG AG		
*End. PEDF*	5176	F: GCT GGC TTT GAG TGG AAC GA	244 nt	58 °C
		R: GTG TCC TGT GGA ATC TGC TG		
*Total PEDF*	5176	F: TAC TGA GGG ACA CAG ACA CA	99 nt	58 °C
		R: CAA TGA TGA TGA TGA TGA TGG		
*RPPH1*	85495	F: AGC TGA GTG CGT CCT GTC ACT	63 nt	58 °C
		R: TCT GGC CCT AGT CTC AGA CCT T		
*CTSD*	1509	F: AGG CAA AGG CTA CAA GCT GTC	459 nt	58 °C
		R: TGT GCT CTG GAT CAG CTC TAC		
*CRALBP*	6017	F: TCA CCA CGA CCT ACA ATG TGG	243 nt	58 °C
		R: AAC TAC AGT TCA GCT GGC AGG		
*SB100X*	Synthetic sequence	F: CTT GCA AGC CGA AGA ACA CC	400 nt	58 °C
		R: CAT TCC TCC TGA CAG AGC TG		
*EGFP/Venus*	Synthetic sequence	F: AGC TGA CCC TGA AGC TGA TCT	328 nt	58 °C
		R: ACG TTG TGG CTG TTG TAG TTG T		

GAPDH—Glycerine aldehyde-3-phosphate dehydrogenase, KRT8—Keratine 8, VEGF—Vascular endothelial cell growth factor, ZO-1—Zonula occludens 1, VEGF-R2—VEGF receptor 2, IRDRL—Inverted repeat/direct repeat left (TIR left), rPEDF—recombinant pigment epithelium-derived factor, PEDF—endogenous PEDF, RPPH1—Ribonuclease P RNA Component H1, CTSD—Cathepsin D, CRALBP—Cellular retinaldehyde-binding protein, SB100X—hyperactive *Sleeping Beauty* transposase, EGFP—enhanced green fluorescent protein, Venus—yellow fluorescent protein, nt—nucleotides.

## Data Availability

Data will be available at Zenodo via the following doi: 10.5281/zenodo.7816967.

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
