# Peer review of "Enhanced Biosafety of the Sleeping Beauty Transposon System by Using mRNA as Source of Transposase to Efficiently and Stably Transfect Retinal Pigment Epithelial Cells"

_biomolecules, 2023, doi:10.3390/biom13040658_

Round 1

Reviewer 1 Report

To date, neovascular age-related macular degeneration treatment consists in regular intravitreal injections of anti-VEGF to minimize the appearance of neovessels, responsible of vision loss. In this article presented by Harmening and co-workers, the use of the Sleeping Beauty transposon system as mRNA to enhance production of PEDF, a natural antagonist to VEGF, in retinal pigment epithelium (RPE) cells is evaluated. The Sleeping Beauty transposon system is usually used with the transposase as plasmid DNA but it may lead to undesirable effects, which can be circumvented with mRNA. Herein, authors test the transposase as an mRNA in ARPE-19 cells by comparing it to the DNA approach. They show stable and long-term expression of reporter gene and PEDF when using the transposase as mRNA.

Despite the interesting results, this article needs to be clarified on some points:

-            The figure citations do not always correspond to what is stated in the manuscript or are sometimes missing which makes it hard to follow.

-            The structure of the manuscript is also not ideal. The major aim of this paper is to validate the use of SB100X-mRNA and pT2-CMV-PEDF/EGFP, first in ARPE-19 cells and then in primary human RPE cells. The first two sections of the manuscript are working in this sense but the third section goes back to testing pT2-CAGGS-Venus and SB100X-DNA and mRNA, which was already done in the first section using the same experiments (by flow cytometry to determine the number of positive cells and mean fluorescence intensity and by immunofluorescence micrographs). Why going back when you already showed that mRNA works? And what is the point of these experiment? It is not clear.

-            The quality of the fluorescence images is quite poor and makes it hard to distinguish the cells.

-            Consistency in figure 2, 3, 4 and 5 legends would also help the reader (SB100X-DNA could be always depicted in black for example, it is a black circle on figure 2 and last mentioned while it is showed as a black dot on figure 3 and mentioned first and it is a white histogram on figure 4).

Figure 2: For several conditions (mRNA 1:1 and mRNA 1:16 starting at 114 and mRNA 1:28 starting at 158 days) images look more like epifluorescence background than actual fluorescence due to transfection. A non-transfected control is clearly missing and in my opinion, a control as simple as DAPI should have been used to help the reader visualize the cells. Furthermore, it would have been interesting to have the same timepoints for each cell line. DNA brightfield images are undistinguishable due to the images quality.

Figure 3: How do you explain the differences between the two western blot experimental series? You do not mention any hypothesis on why in one hand PEDF is detectable up to 500 days in the top blot (consistent with the flow cytometry results) and is undetectable at 277 days (already faint at 165 days) in the bottom blot. The same observations can be made with SB100X-mRNA 1:20 where the top blot results are very different from the bottom blot and the cytometry results. Moreover, in the original blots provided, why did you not test sample n°6 (mRNA 1:28) at 165 days, but did n°7? How did you determine no PEDF secretion as mentioned in the figure legend if you did not perform western blot on these samples?

Figure 4: You mention in the legend that panel A illustrates mRNA expression levels of several endogenous genes but there is SB100X, EGFP/Venus and rPEDF conditions, which are not endogenous. Please revise the organization of the figure or the legend to make it clearer (and cite corresponding panels more often in the text). Moreover, you state (lines 338-339) that SB100X-mRNA + pT2-CMV-PEDF/EGFP at 1:20 will be used for further studies on ARPE-19 cells but this condition was not tested on panels A and B. Why?

The representation chose for the RT-qPCR results is not the best. It would be preferable to show the data of the non-transfected cells rather than only the tested groups relatively to the control group. It would be easier for the reader to understand the figure that way. Also, why did you not test BS100X-DNA in panel C? The histogram legend is missing in panel E.

Figure 5: Same comment as for figure 2 on the quality of the fluorescence images and missing control.

Figure 6: Non-transfected control is missing.

Specific comments

Line 249: There is a reference problem

Lines 292 to 294: Please correct to “means of fluorescence intensity”. Figure 2C does not show the MFI.

Lines 304 to 306: There is still a paragraph from the writing guidelines

Lines 326 to 328: “For 500 days and 277 days, respectively” but there is no mention of which conditions it refers.

Lines 351-352: What do you mean? How did you measure PEDF secretion and where are the data?

Lines 376-377: The data for the non-transfected cells (Figure 4D) and SB100X-DNA (Figure 4C and 4D) which are both controls are missing.

Lines 381-382: The statistical data do not support the “increased significantly” statement (p≥0.05 as mentioned in the figure legend).

Line 383: “was 108-fold higher”. Higher than what?

Lines 447-448: Not clear can you please rephrase.

Line 524: “After 311 respectively 270 days”, please correct

Line 531: Change “APRE-19” to “ARPE-19”

Lines 564-565: Is it really SB100X-mRNA? Please review this sentence.

Lines 569 to 574: From which data do you conclude that “PEDF-transfected cells had a benefit in proliferation and overgrew the non-transfected cells”? No immunofluorescence images were shown for the SB100X-mRNA + pT2-CMV-PEDF/EGFP condition. Paragraph n°6 from the discussion needs to be revised.

Reviewer 2 Report

The reviewer enjoyed reading the manuscript. However, some issues were listed to improve the manuscript. Thanks for checking it. 

Methods: 

2.9. Please solve this issue; Errors! References source not found

Table 1. genes' references (Gene IDs) need to be added in the tabe for matching with the sequence provided in the table.

Results:

Figure 2's ARPE-19's morphology detected with fluorescence seems abnormal. Images for ARPE-19 conditions need to be provided to show that ARPE-19 is healthy or normal.

Figure 3's Missing lanes for western blot seems incorrect. The same western blot membrane has the same lanes for all bands, basically. Thanks for checking and revising them if there needs improvements.

Figure 7's western blot also has the same issue with Figure 3's western blot.

Round 2

Reviewer 1 Report

The reviewer thanks the authors as they took into account most of the comments. However, the presentation of the WB data of Figure 3C is still a bit problematic. Can you please provide in the supplementary data the WB experiments showing that there is no PEDF detected for the conditions marked as * ? Having the original blots is good but the conditions marked as * are not tested in the blots presented. 

For example, the presentation of the results of the SB- condition in the bottom blot is fine as you can see that at 13 days post-transfection there is no more PEDF and then understand that you did not pursue analysis on this sample. All samples should have been presented the same way.

Author Response

Dear reviewer, 

Thank you very much, this comment is highly appreciated. We agree that this is a very good way to show all data without overloading the manuscript. Additionally, two graphs showing the fluorescence analysis of all cultures over the whole observation period were added to complete the dataset, also because some terminations were based on the rPEDF secretion and the percentage of fluorescent cells. To be consistent, a respective figure was added in the supplemental material for the ARPE-19 cells transfected with pT2-CAGGS-Venus (manuscript figure 2) showing the percentage of fluorescence cells for all cultures over the whole observation periods.

Sincerely, 

Nina Harmening